# GTIP: A Gaming-Based Topic Influence Percolation Model for Semantic Overlapping Community Detection

**DOI:** 10.3390/e24091274

**Published:** 2022-09-09

**Authors:** Hailu Yang, Jin Zhang, Xiaoyu Ding, Chen Chen, Lili Wang

**Affiliations:** 1School of Computer Science and Technology, Harbin University of Science and Technology, Harbin 150001, China; 2School of Automatic Control Engineering, Harbin Institute of Petroleum, Harbin 150028, China; 3School of Computer Science and Technology, Chongqing University of Posts and Telecommunications, Chongqing 400065, China

**Keywords:** semantic social networks, community detection, topic influence, percolation mechanics, game theory

## Abstract

Community detection in semantic social networks is a crucial issue in online social network analysis, and has received extensive attention from researchers in various fields. Different conventional methods discover semantic communities based merely on users’ preferences towards global topics, ignoring the influence of topics themselves and the impact of topic propagation in community detection. To better cope with such situations, we propose a Gaming-based Topic Influence Percolation model (GTIP) for semantic overlapping community detection. In our approach, community formation is modeled as a seed expansion process. The seeds are individuals holding high influence topics and the expansion is modeled as a modified percolation process. We use the concept of payoff in game theory to decide whether to allow neighbors to accept the passed topics, which is more in line with the real social environment. We compare GTIP with four traditional (GN, FN, LFM, COPRA) and seven representative (CUT, TURCM, LCTA, ACQ, DEEP, BTLSC, SCE) semantic community detection methods. The results show that our method is closer to ground truth in synthetic networks and has a higher semantic modularity in real networks.

## 1. Introduction

In recent years, with the rapid development of mobile internet technology and the continuous popularization of mobile terminal devices, social platforms such as Micro-blog, WeChat, QQ, SNS, RSS, etc., have changed social interaction deeply. People can join or set up their own community and update their status in the form of text, pictures, and videos to realize the sharing, dissemination, and acquisition of personal information. According to statistics from comScore, Inc. (Reston, VA, USA), as of 2018, an average of 395,833 people logged in to WeChat per minute and 19,444 people were engaged in video or voice chat; Sina Micro-blog sent or forwarded 64,814 microblogs per minute; Facebook users shared an average of four billion dynamic items of information per day; Twitter processed 340 million items of data per day; Tumblr authors published an average of 27,000 new posts per minute; and Instagram users shared an average of 3600 photos per day. Facing this data explosion caused by the growing to social media data, the traditional topological space of social networks is shifting towards a rich semantic form which poses great challenges to the detection of social network communities.

Community detection can effectively improve the performance of social application systems. For example, by analyzing the social behavior patterns of network users and detecting the audience groups of social services, the commercial value of advertising and product marketing can be significantly improved [1]. Han et al. [2] used community detection to realize information transfer between networks and solved the cold start problem of recommendation systems caused by network sparsity. In addition, community detection is widely used in network embedding [3], public health [4], and link prediction [5].

In conventional community detection methods, the network is represented as a topology graph and the nodes do not contain semantic information. Representative methods in this field include the GN (Girvan–Newman) algorithm [6], FN (Fast Newman) algorithm [7], CPM (Cluster Percolation Method) algorithm [8], and Louvain algorithm [9]. In recent research, Qiao et al. [10] proposed Picaso, a parallel community discovery model which uses the Mountain model to calculate the weight of each edge in the network and apply a gradient algorithm to discover the community structure. To solve the problem of community detection in large-scale complex networks, Lu et al. [11] proposed an improved label propagation algorithm using node importance ranking. Lyzinski et al. [12] embedded graphs in Euclidean space to obtain their lower-dimensional representation, then used non-parametric graph reasoning technology to identify the structural similarity between communities. This method performed well in detecting fine-grained community structures. Tagarelli et al. [13] integrated multi-layer network community modularity, which retains multi-layer topology information and optimizes the edge connectivity of multi-relational communities.

In semantic community detection tasks, the nodes are the basic components of the topology graph as well as the carriers of semantic information which leads to fundamental changes in the community’s form [14]. For example, after considering the document attributes of nodes, the common topics between nodes play a decisive role in the formation of the community. Two people who share a common topic may join the same community even if they do not have a strong connection in the topology graph [15]. Therefore, the use of semantic information to analyze the correlation between network nodes has become a critical issue in this field.

The Probabilistic Topic Model (PTM) is a common semantic representation method used for social network nodes [16]. For example, Xin et al. [17] defined the semantic feature of nodes according to the similarity between user documents and a set of global topics, then adopted multi-sampling to accelerate the convergence of the algorithm. He et al. [18] transformed LDA (Latent Dirichlet Allocation) and Markov Random Field (MRF) into a unified factor graph to form an end-to-end learning system for community detection, then derived an effective propagation algorithm to train their parameters. Jin et al. [19] stated that links in the network contain semantic information as well. They proposed a new probabilistic model for link community detection, and developed a dual nested Expectation Maximum (EM) algorithm to learn the model. Wang et al. [20] found that there are correlations between topics which significantly affect community structures. They proposed a Topic Correlations-based Community Detection (TCCD) model which can simultaneously output the community structure and the semantic interpretation of nodes. Node attributes can be used to address semantic data as well; for example, Fang et al. [21] grouped nodes that satisfied both structure cohesiveness and keyword cohesiveness into the same community.

Non-negative Matrix Factorization (NMF) has good performance in discovering implicit patterns from high-dimensional data. Therefore, scholars have integrated semantic information into the adjacency (or feature representation) matrix and used NMF to analyze the correlation between nodes. For example, Pei et al. [22] proposed a clustering framework based on Non-negative Matrix Tri-Factorization (NMTF) which can effectively identify both user similarity and message similarity. Qin et al. [23] introduced an adaptive parameter to control the contribution of the network topology and content information and use NMF to discover semantic communities. Wang et al. [24] set the member matrix and attribute matrix as two groups of parameters of NMF, which allows semantic interpretation for the communities to be added. Yang et al. [25] introduced an adaptive weighted group for sparse low-rank regularization in NMF in order to automatically obtain the number of semantic communities.

Deep learning has a natural advantage in attribute representation of high-dimensional data; thus, researchers have begun to introduce semantic attributes into the feature dimension of deep learning models [26]. For example, Jin et al. [27] proposed a uniformed graph representation of network topology and semantic information and developed a multi-component network embedding approach via a deep autoencoder. Cao et al. [28] designed a combination matrix consisting of a modularity matrix for linkage information and a Markov matrix for content information. After matrix factorization, the matrix is used as the input of the multi-layer deep auto-encoder framework for obtaining the deep representation of the graph. Jin et al. [29] proposed that the words in user documents have a hierarchical structure. They proposed a new Bayesian probability model which can explain the multiplex semantic community more clearly. He et al. [30] developed a co-learning strategy to jointly train the structure and semantic parts of the model by combining a nested EM algorithm and belief propagation.

While the above methods have made a great many exploratory contributions to the field of semantic community detection, there are several remaining deficiencies:(1)When measuring the semantic relevance between nodes, each topic receives the same status without considering the difference of topic influence.(2)There has been little exploration of the impact of topic propagation and influence propagation in community detection.(3)Methods based on deep learning require a large number of samples, high computational performance, and long training times. When the network evolves rapidly, these methods cannot meet the online requirements of social systems.

To better cope with these situations, and inspired by the information dissemination in social networks, we propose a user topic influence propagation model based on percolation theory that uses the Nash equilibrium to generate communities in a game-based way. Experiments with real social networks show that the proposed method has a high semantic modularity [17] in social networks with rich semantic attributes. In addition, the algorithm can converge in a short time without additional training. In summary, the contributions of this paper include:(1)Integrating topic influence into the correlation analysis of nodes, which makes the community detection process conform to the law of information dissemination in social networks.(2)A proposed one-dimensional diffusion model in percolation mechanics that can quantify the propagation of topic influence, which in turn can describe the impact of nodes near the topic source in the semantic space more accurately and solve the situation in which high-influence nodes in the network present a low influence score.(3)Use of the Nash equilibrium from game theory to generate communities, thereby identifying overlapping and non-overlapping communities at the same time and identifying community structures with smaller granularity.

## 2. LDA Model of Semantic Social Networks

### 2.1. LDA Representation of Nodes

The semantic space representation of nodes is generated based on LDA, a three-tier Bayesian probability model used for document-topic generation, including words, topics, and documents. LDA considers documents to be composed of topics, and each topic can be presented with a set of keywords. For example, technology topics have a high probability of containing the keywords: “Chip” and “Artificial Intelligence”. The probability distribution of the document on each topic shows the relevance of the document to each topic. The mathematical symbols involved in LDA are shown in Table 1.

The LDA vector is stored as a triplet, (*w*, *d*, *z*), where $w_{i},d_{i}$ and $z_{i}$ are the number, the node number, and the topic number of keyword *i*, respectively [31]. Figure 1 shows the data storage structure of the LDA vector, in which the shadow part represents the same elements in the vector. For example, $w_{i1}=w_{i2}=w_{i4}=w_{i5}$ indicates that $w_{i1},w_{i2},w_{i4},w_{i5}$ are the same words, $d_{i1}=d_{i3}=d_{i5}=d_{i6}$ indicates that $w_{i1},w_{i3},w_{i5},w_{i6}$ are the keywords of the same node $d_{i1}$, and the keyword $w_{i1}$ appears twice in $d_{i1}$. Additionally, $z_{i1}=z_{i2}=z_{i6}$ indicates that $z_{i1},z_{i2},z_{i6}$ belong to the same topic $z_{i1}$, the keyword $w_{i1}$ appears twice in $z_{i1}$, and $z_{i1}$ belongs to $d_{i1}$ and $d_{i2}$, respectively. According to [31], the mathematical descriptions of w,d,z are as follows:(1)θ∼ Dir (α); the topic distribution θ of nodes follows the Dirichlet distribution (noted as Dir in the formula) with parameter α.(2)$z_{i}|\theta^{\left(d_{i}\right)}∼$ Multinomial $\left(\theta^{\left(d_{i}\right)}\right)$; the probability of topic $z_{i}$ in node $d_{i}$ under topic distribution θ follows Multinomial distribution (noted as Multinomial in the formula).(3)λ∼ Dir (β); the keyword distribution follows the Dirichlet distribution with parameter β.(4)$w_{i}|z_{i},\lambda^{\left(z_{i}\right)}∼$ Multinomial $\left(\lambda^{\left(z_{i}\right)}\right)$, the probability of keyword $w_{i}$ in topic $z_{i}$ under keyword distribution λ follows Multinomial distribution.

To generate the LDA model, the first step is to extract the distribution of keywords that satisfy λ∼ Dir (β). Next, the topic distribution is extracted for each document in the corpus, satisfying θ∼ Dir (α). Finally, for each keyword, topics and keywords are further extracted to satisfy $z_{i}|\theta^{\left(d_{i}\right)}∼$ Multinomial $\left(\theta^{\left(d_{i}\right)}\right)$ and $w_{i}|z_{i},\lambda^{\left(z_{i}\right)}∼$ Multinomial $\left(\lambda^{\left(z_{i}\right)}\right)$, respectively.

**Figure 1 entropy-24-01274-f001:**
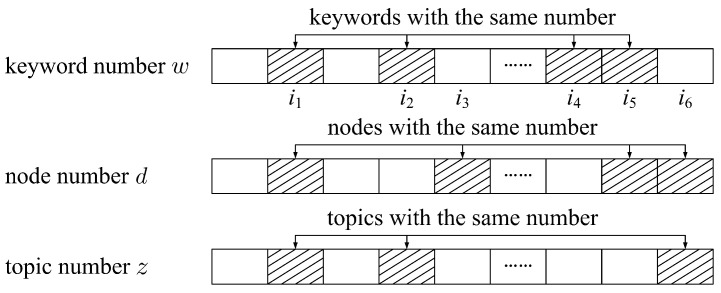
Data storage structure of LDA vector.

**Table 1 entropy-24-01274-t001:** Description of notation.

Notation	Description
*G*	Semantic social network
|G|	The number of nodes in *G*
*N*	The total number of the keywords in *G*
$N_{i}$	The number of keywords of node $G_{i}$
*w*	Keyword vector
$w_{i}$	The *i*-th keyword in vector *w*
*d*	Node number vector corresponding to *w*
$d_{i}$	The node number to which $w_{i}$ belongs
*z*	Topic number vector corresponding to *w*
$z_{i}$	The topic number to which $w_{i}$ belongs
$\theta^{\left(g_{i}\right)}$	The topic distribution probability of node *i*
$\lambda^{\left(j\right)}$	The distribution of keywords in topic *j*
$\lambda_{w_{i}}^{\left(j\right)}$	The probability that $w_{i}$ belongs to topic *j*
α	Prior parameter of topic distribution for each node
β	Prior parameter of keyword distribution within a topic

**Table 2 entropy-24-01274-t002:** Differences between fluid percolation and semantic percolation.

Attribute	Fluid Percolation	Semantic Percolation
Percolation area	Adjacent area	Adjacent nodes
The percolation process	Reversible	Irreversible
Percolation direction	Flow to percolation area	From high Influence nodes to low Influence nodes
Percolation condition	Contains fluid	Determined by the game

### 2.2. Gibbs Iterative Process

In statistics, Gibbs sampling is a Markov Monte Carlo (MCMC) algorithm which is used to approximately extract sample sequence from a multivariate probability distribution when it is difficult to directly sample. The key is to establish a posterior estimate for a sample and perform Gibbs sampling on the posterior estimate expression.

The expression of the Bayesian relation of *z* and *w* is
(1)$$\begin{array}{cc} \; & P\left(z_{i}=j|w_{i}\right)P\left(w_{j}\right)=P\left(w_{i}|z_{i}=j\right)P\left(z_{i}=j\right)\\ \; & \Rightarrow P\left(w_{i}\right)=\sum_{j=1}^{\left|z\right|}P\left(w_{i}|z_{i}=j\right)P\left(z_{i}=j\right) \end{array}$$

After transformation, we have
(2)$$\begin{array}{cc} \; & P\left(z_{i}=j|z_{-i},w_{i}\right)P\left(w_{j},w_{-i}\right)=\\ \; & P\left(w_{i}|z_{i}=j,z_{-i},w_{-i}\right)P\left(z_{i}=j|z_{-i}\right) \end{array}$$
(3)$$P\left(z_{i}=j|z_{-i},w_{i}\right)\propto P\left(w_{i}|z_{i}=j,z_{-i},w_{i}\right)P\left(z_{i}=j|z_{-i}\right)$$

The process of Gibbs sampling is as follows:

(1) $z_{i}$ is initialized as a random integer between 1 and *K* (i=[1,2,...,N]), which is the initial state of the Markov chain.

(2) According to the literature [32], the right side of Equation (Equation 3) can be expanded as



(4)
$$P\left(w_{i}|z_{i}=j,z_{-i},w_{-i}\right)=\frac{f_{-i,j}^{\left(w_{i}\right)}+\beta }{f_{-i,j}^{\left(\cdot \right)}+\left|w\right|\beta }$$


(5)
$$\begin{array}{cc} P(z_{i}=j|z_{-i}) & =\int P\left(z_{i}=j|\theta^{\left(d_{i}\right)}\right)P\left(\theta^{\left(d_{i}\right)}|z_{-i}\right)d\theta^{\left(d_{i}\right)}\\ \; & =\frac{f_{-i,j}^{\left(d_{i}\right)}+\alpha }{f_{-i,\cdot }^{\left(d_{i}\right)}+\left|z\right|\alpha } \end{array}$$



Therefore, we have
(6)$$P\left(z_{i}=j|z_{-i},w_{i}\right)\propto \frac{f_{-i,j}^{\left(w_{i}\right)}+\beta }{f_{-i,j}^{\left(\cdot \right)}+\left|w\right|\beta }\frac{f_{-i,j}^{\left(d_{i}\right)}+\alpha }{f_{-i,\cdot }^{\left(d_{i}\right)}+\left|z\right|\alpha }$$

In Equation (Equation 6), |w| and |z| denote the number of keywords and topics, respectively, $f_{-i,j}^{\left(w_{i}\right)}$ represents the number of words assigned to topic *j* that are the same as $w_{i}$, $f_{-i,j}^{\left(\cdot \right)}$ represents the number of words assigned to topic *j*, $f_{-i,j}^{\left(d_{i}\right)}$ represents the number of words assigned to topic *j* in node $d_{i}$, $f_{-i,\cdot }^{\left(d_{i}\right)}$ represents the number of all the words assigned to a topic in node $d_{i}$, and $z_{i}$ is updated iteratively according to Equation (Equation 6).

(3) When step (2) has iterated enough times (when $P(z_{i}=j|z_{-i},w_{i})$ converges), the process ends. We now normalize $P(z_{i}=j|z_{-i},w_{i})$ to obtain the keyword topic probability matrix $B_{i,j}$, $B_{i,j}=P\left(z_{i}=j|w=i\right)$, $B_{i,\cdot }=1$.

### 2.3. Semantic Feature Representation of Nodes

In a semantic social network G=(V,E,T), the node set *V* represents the users in the semantic social network, the edge set *E* represents the relationship between users, and *T* is the document collection, representing the text information published by users.

We used Gensim (a topic generation toolkit in Python) to extract *K* topics in *T* as the base of a *K*-dimensional semantic space. The coordinate $m_{i}$ of the node $v_{i}$ ($v_{i}\in V$) in the semantic space can be expressed by the mean value of the keywords in the document $t_{i}$ ($t_{i}\in T$) published by $v_{i}$, which is shown in Equation (Equation 7).
(7)$$m_{i}=\frac{\sum_{j=1}^{N_{i}}B_{N_{i,j},\cdot }}{N_{i}}$$

In Equation (Equation 7), $N_{i}$ represents the number of keywords (the words with the highest cosine similarity to the topic that $t_{i}$ belongs to) in document $t_{i}$, $N_{i,j}$ represents the *j*-th keyword in document $t_{i}$, and $B_{N_{i,j},\cdot }$ represents the coordinate (expressed as the sequence of the cosine similarity between the *j*-th keyword and *K* topics) of the *j*-th keyword in document $t_{i}$ in the *K*-dimensional semantic space.

## 3. Modeling Topic Influence Based on Percolation Mechanics

### 3.1. Motivation

The flow of a fluid through porous media (soil voids or other permeable media) is called percolation. Each percolation source point contains a certain amount of substance, which diffuses to the area in a finite space that has not been penetrated. In the example shown in Figure 2, the grid represents the percolation area. We assume that there are three percolation source points in the figure, labeled red, blue, and green here. In real percolation process, percolation occurs when the difference between the source point and the adjacent area reaches a threshold, which is measured by the point source function. In this example, we simply assume that the probability of percolation is 50%. After four infiltrations, the percolation state changes from Figure 2a,b.

It can be found that from the three source points the substance gradually penetrates into the adjacent areas. Inspired by this, we propose to construct the semantic social network topic percolation equation using percolation theory. Our motivation stems from the following four perspectives. First, both fluid percolation and semantic percolation need to be adjacent to the infiltration area. Second, similar to fluid percolation, in semantic social networks, whether users receive topics from neighbors (i.e., semantic percolation) is subject to a threshold, which in this paper is measured by the payoff concept from game theory. Next, both fluid percolation and semantic percolation are multiple source points percolating simultaneously, and this property can be simulated for community detection using a seed expansion strategy. Finally, all source points have the same status, which avoids the problem that nodes with less local influence cannot expand and promotes the formation of local communities. The differences between fluid percolation and semantic percolation are shown in Table 2.

### 3.2. Modeling Topic Influence

In this section, we construct the topic percolation differential equation; the symbols used are provided in Table 3. We propose topic influence percolation strength to measure the capacity of topics to influence the percolation area. In our model, each node is a fixed-size solid sphere filled with unequal topic influence in the semantic space. In the model, *S* has a virtual dimension $\left[\lambda \gamma^{-1}\right]$. In the semantic space, the inner product $m_{i}\cdot m_{j}$ represents the semantic correlation between nodes $v_{i}$ and $v_{i}$. The more similar the semantic coordinates of $v_{i}$ and $v_{i}$ are, the larger $m_{i}\cdot m_{j}$ is. We define $Z_{i\to j}=1/m_{i}\cdot m_{j}$ to represent the topic propagation space coordinate of node $v_{j}$ with node $v_{i}$ as the source point, which satisfies $Z_{i\to i}=0$, and $Z_{i\to j}\to \infty$ when $m_{i}\cdot m_{j}\to 0$.

We design three rules to construct the percolation dynamics of topic influence, based on which the second-order partial differential equation of topic percolation *Z* is provided in Equation (Equation 8)

(1) The topic influence of a percolation source point is greatest at the initial state, and spreads outward with the percolation of topic influence.

(2) As the topic influence of the source point continuously penetrates into the surrounding area, the influence of the source point on other nodes becomes smaller.

(3) While the nodes under the influence of the source point absorb and weaken the topic influence of the source point, the influence of the topic contained in the source point is enhanced.



(8)
$$\frac{\partial^{2}S}{\partial Z^{2}}=\frac{1}{\eta_{z}}\frac{\partial S}{\partial D}$$



The initial condition of Equation (Equation 8) is as follows:(9)$$S\left(Z,0\right)=\kappa_{0}\delta \left(Z\right)$$

Here, δ(Z) is a Dirac function, which satisfies the requirement that the value of the function (except source point *a*) be equal to 0 and the integral over the entire domain equal to 1. The expression of δ(Z) is
(10)$$\left\{\begin{array}{ccc} \delta (Z)=\delta (Z-a) & , & x\neq a,\\ \int_{-\infty }^{+\infty }\delta (Z)dZ=1 & , & x=a \end{array}\right.$$

Here, S(Z,0) denotes the topic influence percolation strength when the distance between the source point and the affected node is 0. At this point, the influence is concentrated on the source point, $S\left(Z,0\right)=\kappa_{0}$.

The boundary conditions of Equation (Equation 8) are as follows:(11)$$\left\{\begin{array}{c} S(\infty ,D)=0\\ \frac{\partial S(\infty ,D)}{\partial Z}=0 \end{array}\right.$$

Equation (Equation 11) indicates that *S* and the partial differential of *S* with respect to *Z* becomes 0 when Z→∞.

Because the partial differential equation is established using physical phenomena, we use Dimensional Analysis (DA) to solve Equation (Equation 9). The basic principle of DA is Buckingham π theorem. The theorem states that if the formula of a physical process contains *n* physical quantities and *k* of them have independent dimensions, then the formula can be transformed into an equivalent function containing n−k dimensionless numbers $\pi_{i}$ composed of these physical quantities.

The topic influence percolation strength *S* is a function of κ, *z*, *D* and $\eta_{z}$. Suppose that $F(S,\kappa ,Z,D,\eta_{z})=0$; then, the dimension of *S* and κ is $\left[\lambda \gamma^{-1}\right]$ and [λ], respectively, and *S* is proportional to $\lambda \;/\sqrt{\eta_{z}D}$. Using Buckingham π theorem and selecting $S,D,\eta_{z}$ as the basic variable, we have
(12)$$F(\frac{\kappa }{S\sqrt{\eta_{z}D}},\frac{Z}{\sqrt{\eta_{z}D}})=0$$
(13)$$\frac{\sqrt{4\pi \eta_{z}d}}{\kappa }S\left(Z,d\right)=f\left(\frac{Z}{\sqrt{4\eta_{z}d}}\right)$$

Next, we determine the undetermined function *f*. Let variable $\psi =Z/\sqrt{4\eta_{z}D}$; then,
(14)$$S\left(Z,D\right)=f\left(\psi \right)\frac{\kappa }{\sqrt{4\pi \eta_{z}D}}$$

Combined with Equation (Equation 8), we have
(15)$$\frac{d}{d\psi }\left(\frac{df}{d\psi }+2\psi f\right)=0$$

The boundary conditions of Equation (Equation 11) becomes
(16)$$\left\{\begin{array}{c} f(\infty )=0\\ \frac{df(\infty )}{d\psi }=0 \end{array}\right.$$

After simplification, we have
(17)$$\frac{df}{d\psi }+2\psi f=c$$

Here, *c* is a constant. By substituting Equation (Equation 8) into Equation (Equation 17), we have c=0; therefore, the general solution of Equation (Equation 17) is $f=\omega_{o}e^{-\psi^{2}}$. According to the hypothesis, the topic influence of the source point is conserved; therefore,
(18)$$\int_{-\infty }^{+\infty }SdZ=\kappa \int_{-\infty }^{+\infty }e^{-u}du=\sqrt{\pi }$$

As $\int_{-\infty }^{+\infty }e^{-u}du=\sqrt{\pi }$, $\omega_{0}=1$, therefore,
(19)$$S\left(Z,D\right)=\frac{\kappa }{\sqrt{4\pi \eta_{z}D}}\exp\left\{-\frac{Z^{2}}{4\eta_{z}D}\right\}$$

After the transposition of terms, we have
(20)$$\frac{S(Z,D)}{\kappa }=\frac{1}{\sqrt{2\pi }\sqrt{2\eta_{z}D}}\exp\left\{-\frac{Z^{2}}{2{\left(\sqrt{2\eta_{z}D}\right)}^{2}}\right\}$$

Equation (Equation 20) is a typical standard normal function with the topic propagation space coordinate *Z* as the horizontal axis and the topic influence percolation strength *S* as the vertical axis. According to the mathematical properties of the standard normal function, the instantaneous influence of the source point follows a normal distribution along the *Z* direction at any *D* point in the strength field in one-dimensional unbounded semantic space. With increasing distance *D*, the peak value of influence strength decreases while the range of affected nodes becomes wider, and the distribution curve tends to become more stable.

According to the 3σ principle, the probability of topic influence of each node outside (μ+3σ,μ−3σ) is less than 0.3%. Therefore, μ−3σ<Z≤μ+3σ can be regarded as the actual range of random variable *Z*, and the topic influence of nodes is only valid within the range of $3\sigma =3\sqrt{2\eta_{z}D}$.

## 4. The Game Process of Topic Influence Percolation

In social networks, each individual has free will and can decide whether to join a community after weighing the advantages and disadvantages, which is consistent with the behavior of the players in game theory. In semantic social networks, users influence people around them with their preferred topics and are influenced in turn by the topics held by others. When affected by different topics, people react differently. For high-impact topics that they prefer and are hotly discussed by the public, they continue to track the progress of these topics and further spread them. On the contrary, they do not pay further attention. From the perspective of game theory, all social individuals are considered to be rational and selfish players and follow certain rules to join the semantic community with greater influence and closer to their preferred topics in order to maximize their payoffs and achieve Nash equilibrium.

### 4.1. Basic Elements

The basic elements of our game model are as follows.

(1) Players: all nodes except the seed nodes (unequilibrium nodes) in semantic social networks.

(2) Strategy $P_{i}$: each player chooses a single strategy; $P_{i}=1$ ($P_{i}=0$) means that after being affected by the topic, node $v_{i}$ does (does not) spread the topic and joins (refuses to join) the community to which the topic belongs.

(3) Payoff $U_{i}$: in the percolation dilemma game model, the payoff of node $v_{i}$ is defined as follows:(21)$$U_{i}\left(P_{i},P_{j}\right)=S_{ji}-\xi$$

Here, $U_{i}\left(P_{i},P_{j}\right)$ represents the payoffs of $v_{i}$ of spreading topics from $v_{j}$, $S_{ji}$ represents the percolation strength of the topic from $v_{j}$ to $v_{i}$, and ξ represents the topic percolation loss. The correlation between $P_{i}$ and $U_{i}$ is as follows.:(22)$$P_{i}=\left\{\begin{array}{ccccc} 0 & , & if & \; & U_{i}\left(P_{i},P_{j}\right)\leq 0,\\ 1 & , & if & \; & U_{i}\left(P_{i},P_{j}\right)>0. \end{array}\right.$$

In a semantic social network, if there is a node with greater topic influence than node $v_{i}$ in the percolation area, $v_{i}$ is percolated by topic influence, and the percolation with smaller strength is covered by percolation with higher strength. On the contrary, the influence percolation strength $S_{i}$ of node $v_{i}$ in this area is considered infinite. $S_{i}$ is defined as follows:(23)$$S_{i}=\left\{\begin{array}{ccc} max\{S_{ji}\},j\in G & , & \kappa_{(i)0}<\kappa_{(j)0},\\ +\infty & , & \kappa_{(i)0}>\kappa_{(j)0}. \end{array}\right.$$

In this way, it is only necessary to calculate the payoffs of spreading the topic of nodes that can percolate $v_{i}$, instead of calculating the payoffs of the global nodes. To calculate faster, the topic influence percolation strength *S* is stored in a large root heap.

In Equation (Equation 23), the nodes only propagate one topic and join one community. However, communities in real semantic social networks generally overlap. If joining multiple communities can increase payoffs, players join multiple communities. Joining multiple communities results in a loss of payoffs. For semantic overlapping communities, the payoff is defined as follows:(24)$$\begin{array}{cc} \; & U_{G}\left(i\right)=\sum_{j\in G}U_{i}\left(P_{i},P_{j}\right)-\zeta \left(|R(i)|-1\right)\\ \; & \Rightarrow \zeta =\frac{1}{\left|R(i)\right|}\sum_{j\in G}U_{i}\left(P_{i},P_{j}\right) \end{array}$$

Here, ζ is the loss factor, |R(i)| represents the number of different topics spread by node $v_{i}$, and $U_{i}\left(P_{i},P_{j}\right)$ represents the payoffs of $v_{i}$ spreading only one topic. Obviously, spreading more topics results in the loss of ζ.

Players pursue the maximization of payoffs as well as the maximization of efficiency. In generally, the payoff of joining multiple communities is higher than that of joining a small number of communities; in certain cases, joining a small number of high payoff communities can obtain the equivalent payoffs of joining a large number of low-payoff communities. To maximize the payoff and efficiency at the same time, we define a payoff satisfaction function $\rho_{\left(i\right)}$, which is
(25)$$\rho_{\left(i\right)}=\left\{\begin{array}{cccc} \frac{1}{N}\sum_{k=1}^{N}\sum_{j\in G,j\neq i}U_{k}\left(P_{k},P_{j}\right), & if & \; & N_{i}>1,\\ \frac{1}{2}U_{i}\left(P_{i},P_{j}\right), & if & \; & N_{i}=1. \end{array}\right.$$

Here, $N_{i}$ represents the number of communities that node $v_{i}$ has joined. When $N_{i}=1$, $\rho_{\left(i\right)}$ is set as $U_{i}/2$ to avoid that the initial payoff satisfaction of node $v_{i}$ is too large to join other communities. When N>1, the payoff satisfaction is the average of the payoff function. If $U_{G}\left(i\right)<\rho_{\left(i\right)}$, this means that joining the new community results in decreased payoff. In this case, $v_{i}$ chooses strategy $P_{i}=0$.

### 4.2. Slecting the Source Point

Random selection of the source point may result in percolation failure due to the low influence of the selected node and cause additional time cost. Based on the PageRank algorithm, a source point selection algorithm for topic influence maximization is proposed.

(1) Initialize seedSet, hashMap, and $outlink[v_{i}]$, where seedSet stores the ranked topic influence, hashMap stores the feature pairs (nodeID and topic
influence), and $outlink[v_{i}]$ is an array that stores the pointing nodes of $v_{i}$.

(2) According to different transfer probabilities, the node percolates its influence to the pointing nodes. We construct the following transfer matrix

(26)$$P_{i,j}=\left\{\begin{array}{cc} \frac{M(i,j)}{\sum_{v_{k}\in outlink\left[v_{i}\right]}M\left(i,k\right)}, & outlink[v_{i}]\neq 0,\\ M(i,j)=0, & others. \end{array}\right.$$
to represent the probability of influence passing from $v_{j}$ to $v_{i}$, where M(i,j) is a weighted adjacent matrix with the formula shown in Equation (Equation 27).
(27)$$M\left(i,j\right)=\left\{\begin{array}{cc} m_{i}\cdot m_{j}, & i\to j,\\ 0, & others. \end{array}\right.$$

If node $v_{i}$ points to node $v_{j}$, the edge weight of arc (i,j) is $m_{i}\cdot m_{j}$; otherwise, the edge weight is 0.

(3) The influence of each node depends on the influence of the nodes that point to it. In the iteration process, we use vector vec to store the influence score of each node, which is updated based on Equation (Equation 28).



(28)
$$\alpha P^{T}vec+\left(1-\alpha \right)\frac{\tau }{N}\to vec,\tau ={\left(1,1\cdot \cdot \cdot 1\right)}^{T}$$



Here, α is the damping factor, which is used to prevent excessive influence of nodes, while τ/N is the self-restart vector, which establishes the transition probability for the node pair that does not have direct link. Equation (Equation 28) is repeated until the entire network converges.

(4) We define conversion coefficient ε and multiply the influence score of each node by ε to obtain the topic influence κ, then update hashMap and seedSet. The pseudo-code of the ranking procedure is provided in Algorithm 1.
**Algorithm 1** Slecting SeedSet.**Input:** Network G=<V,E,T>
**Output:**seedSet,hashmap1: 0→seedSet, 0→hashMap;2: Initialize $outlink[v_{i}]$, $v_{i}\in V$;3: Construct M(i,j) and $P_{i,j}$ using Equations (Equation 27) and (Equation 26);4: **while** (not converged)5:     **for** $v_{i}\in V$ **do**6:         Update the influence score based on Equation (Equation 28);7:     **end for**8: **end while**9: Ranking vec→seedSet;10: Feature pairs of vec→hashMap;


### 4.3. Game Rules for Overlapping Community Detection

Based on the topic influence percolation, we propose a game algorithm for overlapping community detection.

(1) A strategy combination is considered to be in Nash equilibrium if no player can increase their payoff by changing decisions unilaterally. In the initial stage, the nodes in the semantic social network are isolated, no payoff is generated, and all local communities are in a state of unequilibrium.

(2) The percolation is a local movement; therefore, choosing a reasonable propagation range (hops) can ensure the effectiveness of the influence and the fast convergence of the algorithm. According to the 3σ principle of Equation (Equation 20), the topic propagation space coordinate *Z* satisfies
(29)$$\mu -3\sqrt{2\eta_{z}D}<Z\leq \mu +3\sqrt{2\eta_{z}D}$$

Here, $Z_{i\to j}=1/m_{i}\cdot m_{j}$, $m_{i}\cdot m_{j}\in \left(0,1\right)$. When $m_{i}\cdot m_{j}=0.2$, ${\left|Z\right|}_{max}=3\sqrt{2\eta_{z}D}=5$, $D_{max}=3$ (after rounding). The experiments in Section 5.3.1 show that the community quality decreases rapidly when $D_{max}>3$. Therefore, to speed up the algorithm, we assume that there is no percolation between $v_{i}$ and $v_{j}$ when $m_{i}\cdot m_{j}<0.2$.

(3) Select nodes sequentially from the head of seedSet; if the node is marked as “divided” in hashMap, select new nodes from seedSet until the node is marked as “not divided”, making it the source point of the percolation.

(4) For $v_{i}$ within three hops of source point $v_{j}$, if $v_{i}$ does not join any community, calculate the non-overlapping payoff function $U_{i}\left(P_{i},P_{j}\right)$. If $U_{i}\left(P_{i},P_{j}\right)>0$, then $v_{i}$ joins $v_{j}$ community and marks $v_{i}$ as “divided” in hashMap, the number of hashMap elements minus 1. If $U_{i}\left(P_{i},P_{j}\right)<0$, skip $v_{i}$ and analyze the next node.

(5) If $v_{i}$ has joined a community and is not in the same community as $v_{j}$, calculate the cosine similarity between $v_{j}$ and the source point of $v_{i}$ community; the expression is as follows:(30)$$\begin{array}{cc} sim(m_{seed(i)},m_{j}) & =\frac{m_{seed(i)}\cdot m_{j}}{|m_{seed(i)}\left|\right|m_{j}|}\\ \; & =\frac{\sum_{g=1}^{k}m_{seed(i)}m_{j,g}}{\sqrt{\sum_{g=1}^{k}m_{seed(i),g}^{2}\sum_{g=1}^{k}m_{j,g}^{2}}} \end{array}$$

Here, we use $\varsigma (v_{i})$ to represent the community collection of $v_{i}$ if $sim(m_{seed(i)},m_{j})>0.8$, merging $\varsigma (v_{i})$ and $\varsigma (v_{j})$. If $sim(m_{seed(i)},m_{j})\leq 0.8$ and the payoff is greater than the payoff satisfaction ($U_{G}\left(i\right)\geq \rho \left(i\right)$), we add $v_{i}$ to $v_{j}$’s community; otherwise, skip $v_{i}$ and find the next node.

(6) When performing an optimal strategy can improve the payoff, the node acts to achieve local Nash equilibrium. Next, we select nodes from the seedSet to play the game until the whole network reaches Nash equilibrium.

(7) When the seedSet is empty and there are elements marked "not divided" in the hashMap, we can accelerate the convergence of the algorithm by randomly assigning these elements to the nearest community.

(8) Nodes affected by the same source point and meeting the game conditions are assigned to the same community, and the semantic community $\varsigma =\varsigma_{1},\varsigma_{2},...,\varsigma_{N}$ is output. The pseudo-code is shown in Algorithm 2.

### 4.4. A Practical Case

Figure 3a shows a directed weighted network $G_{a}$ with six nodes $v_{1},v_{2},...,v_{6}$ where the direction of the edge points to the source of percolation and the weight of the edge represent the difficulty of topic influence percolation.

According to Equations (Equation 26) and (Equation 27), the weighted adjacent matrix of $G_{a}$ is
(31)$$M\left(i,j\right)=\left(\begin{array}{cccccc} 3 & 0 & 0 & 0 & 0 & 0\\ 0 & 0 & 0 & 0 & 0 & 0\\ 1 & 4 & 0 & 0 & 1 & 2\\ 0 & 0 & 2 & 0 & 0 & 4\\ 0 & 0 & 0 & 1 & 0 & 3\\ 0 & 0 & 0 & 0 & 0 & 0 \end{array}\right)$$
and the transfer matrix of $G_{a}$ is
(32)$$P\left(i,j\right)=\left(\begin{array}{cccccc} 1 & 0 & 0 & 0 & 0 & 0\\ 0 & 0 & 0 & 0 & 0 & 0\\ 1/8 & 1/2 & 0 & 0 & 1/8 & 1/4\\ 0 & 0 & 1/3 & 0 & 0 & 2/3\\ 0 & 0 & 0 & 1/4 & 0 & 3/4\\ 0 & 0 & 0 & 0 & 0 & 0 \end{array}\right)$$

**Algorithm 2 **GTIP Algorithm.**Input:** Network G=<V,E,T>,seedSet,hashMap.**Output:** Divided communities $\varsigma =\varsigma_{1},\varsigma_{2},...,\varsigma_{N}$1: **while**
seedSet≠∅2:     *j* = seedSet.top();3:     seedSet.pop();4:     **if** hashMap[j]==false **then**5:         repeat step 2 and step 3;6:     **for** all nodes $v_{i}$ within 3-hops of seed node $v_{j}$ **do**7:         **if** $|\varsigma (v_{i})|=1$ **then**8:             **if** payoff $U_{i}\left(P_{i},P_{j}\right)>0$ **then**9:                 $\pi_{k}\leftarrow v_{i}$, $|\varsigma (v_{i})|++$;10:                 hashMap[i]←false;11:                 hashMap.count−−;12:             **else**13:                 continue;14:             **end if**15:         **else if** $\varsigma (v_{i})\neq \emptyset$ and $\varsigma \left(v_{i}\right)\cap \varsigma \left(v_{j}\right)=\emptyset$ **then**16:             **if** $sim(m_{seed(i)},m_{j})>0.8$ **then**17:                 merging community $\varsigma (v_{i})$ and $\varsigma (v_{j})$;18:             **else**19:                 **if** $U_{G}\left(i\right)>0$ **then**20:                     $\varsigma_{k}\leftarrow v_{i}$;21:                     hashMap[i]←false;22:                     hashMap.count−−;23:                 **else**24:                     continue;25:                 **end if**26:             **end if**27:         **end if**28:     **end for**29: **end while**30: **while**
hashMap.count>031:     $hashMap\left[k\right]\to \varsigma_{k}$;32: **end while**33: return $\varsigma_{1},\varsigma_{2},...,\varsigma_{N}$


The topic propagation space coordinate $Z_{i\to j}=1/m_{i}\cdot m_{j}$; therefore, the coordinate matrix of $G_{a}$ is
(33)$$Z_{i,j}=\left(\begin{array}{cccccc} 0 & 3 & 5 & 2 & 2/5 & 1\\ 3 & 0 & 1 & 1/2 & 2/3 & 1\\ 5 & 1 & 0 & 0 & 1 & 1/2\\ 2 & 1/2 & 0 & 0 & 1 & 1/4\\ 2/5 & 2/3 & 1 & 1 & 0 & 1/3\\ 1 & 1 & 1/2 & 1/4 & 1/3 & 0 \end{array}\right)$$

The topic influence score of the nodes in $G_{a}$ is shown in Table 4.

First, the most influential node $v_{6}$ in Table 4 is selected as the source point of percolation. Due to the small amount of data, we assume that the influence range of the topic is one hop, i.e., d=1.

The nodes affected by $v_{6}$ include $v_{3}$, $v_{4}$ and $v_{5}$. For $v_{3}$, it is affected by $v_{6}$, $v_{2}$, and $v_{5}$. Let the percolation coefficient $\eta_{z}=0.5$ and the dimensionless number π=3. According to Equation (Equation 19), the percolation strength of $v_{6}$, $v_{2}$, and $v_{5}$ to $v_{3}$ are $S_{6,3}=11.60\times \exp\{-0.125\}=10.237$, $S_{2,3}=10.38\times \exp\{-0.5\}=6.296$, and $S_{5,3}=4.70\times \exp\{-0.5\}=2.849$, respectively. Therefore, the node with the greatest influence on $v_{3}$ is $v_{6}$. Assuming that the cost of propagating topics to $v_{3}$ is the topic influence of $v_{3}$ itself, therefore, $U_{6}\left(P_{6},P_{3}\right)>0$, and $v_{3}$ accepts and continues to spread the topic of $v_{6}$ and joins $v_{6}$ community. Similarly, $v_{4}$ and $v_{5}$ are divided into $v_{6}$ community.

The local area covered by the influence of $v_{6}$ reaches Nash equilibrium. Next, $v_{2}$ is selected as the source point of percolation. The influence of $v_{2}$ covers $v_{1}$ and $v_{3}$; $v_{3}$ is marked as “divided”, therefore, we need to compare the topic similarity between $v_{2}$ and the source point of $v_{3}$ community (i.e., $v_{6}$) according to Equation (Equation 30). Suppose that $m_{2}\cdot m_{6}=1$, $|m_{2}|=2$, $|m_{6}|=1$; then, we have $U\left(m_{2},m_{6}\right)=m_{2}\cdot m_{6}/|m_{2}\left|\right|m_{6}|=0.5$. Thus, $U(m_{2},m_{6})<0.8$, the communities of $v_{2}$ and $v_{6}$, are not merged. According to Equations (Equation 24) and (Equation 25), the payoff and payoff satisfaction of $v_{3}$ are $U_{G}\left(3\right)=10.237+6.296-8.267=8.266$ and $\rho_{\left(3\right)}=5.119$, respectively. $U_{G}\left(3\right)>\rho_{\left(3\right)}$; thus, $v_{3}$ joins $v_{2}$ community, forming an overlapping structure. Similarly, we can calculate the topic influence of $v_{2}$ on $v_{1}$ to make the local region reach Nash equilibrium. The community detection result of $G_{a}$ is shown in Figure 3b.

## 5. Experimental Results and Analysis

### 5.1. Experimental Settings

#### 5.1.1. Experimental Environment

All experiments in this paper were performed on a computer with an Intel (R) Core (TM) i5-7500 CPU, 3.40 GHz, and Yuzhan 16GB DDR4 RAM. All the proposed and compared algorithms were programmed in Python.

#### 5.1.2. Compared Algorithms

For complex networks, GTIP was compared to four traditional community detection algorithms: GN (Girvan Newman) [6], FN (Fast GN) [7], LFM (Lancichinetti Fortunato Method) [33], and COPRA (Community Overlap Propagation Algorithm) [34]. GN and FN are non-overlapping community detection algorithms, while LFM and COPRA are overlapping community detection algorithms.

For semantic networks, GTIP was compared to seven semantic community detection algorithms: CUT (Community User Topic) [35], TURCM (Topic User Recipient Community Models) [36], LCTA (Latent Community Topic Analysis) [37], ACQ (Attributed Community Query) [21], DEEP (Deep Learning Method) [28], BTLSC (Background and Two-Level Semantic Community) [29], and SCE (Single Chromosome Evolutionary) [14]. CUT, TURCM, and LCTA generate communities based on Topic Probability Model; ACQ is an attribute graph community detection method; DEEP and BTLSC are both Deep Learning-based semantic community detection methods; and SCE is a new semantic community detection method based on Single-Chromosome Evolutionary.

#### 5.1.3. Evaluation Criteria

Shen et al. [38] introduced Extension *Q*-modularity (EQ) to evaluate the quality of algorithms for identifying highly clustered communities; it is defined as follows:(34)$$EQ=\frac{1}{M}\sum_{t}\sum_{i\in C_{t},j\in C_{t}}\frac{1}{O_{i}O_{j}}\left[A_{i,j}-\frac{K_{i}K_{j}}{M}\right]$$
where $K_{i}$ is the degree of node $v_{i}$, $M=\sum_{i}^{j}A_{ij}$ is the total degree of the network nodes, $A_{i,j}$ is the adjacent matrix of the network, and $O_{i}$ is the number of communities to which $v_{i}$ belongs.

In a semantic social network, the community structure should satisfy both the link density and semantic cohesion between nodes. Xin et al. [17] introduced Semantic *Q*-modularity (SQ) to evaluate the semantic cohesion of the community structure, which is defined as follows:(35)$$SQ=\frac{1}{M}\sum_{t}\sum_{i\in C_{t},j\in C_{t}}\frac{sim(m_{i},m_{j})}{O_{i}O_{j}}\left[A_{i,j}-\frac{K_{i}K_{j}}{M}\right]$$

In Equation (Equation 35), $m_{i}$ and $m_{j}$ is the coordinate of node $v_{i}$ and node $v_{j}$ in semantic feature space, $sim(m_{i},m_{j})$ is the cosine similarity between $v_{i}$ and $v_{j}$ (Equation (Equation 30)), and the range of EQ and SQ is (0,1); the closer this value is to 1, the higher the quality of the community.

Lancichinetti et al. [33] introduced Normalized Mutual Information (NMI) to compare the similarity between the ground truth and the detected communities. The normalized mutual information between partition $C_{X}$ and $C_{Y}$ is defined as follows:(36)$$NMI=1-\frac{1}{2}\left(\frac{H(C_{X}\left|C_{Y}\right.)}{H(C_{X})}+\frac{H(C_{Y}\left|C_{X}\right.)}{H(C_{Y})}\right)$$
where $H(C_{X})$ is the entropy of $C_{X}$ and $H(C_{X}\left|C_{Y}\right.)$ is the variation of information between $C_{X}$ and $C_{Y}$. In the experiments, NMI is used to compare the communities discovered by the algorithm with the ground-truth communities in the artificial network.

### 5.2. Datasets

#### 5.2.1. Artificial Networks

For our experiments, we produced ten artificial networks with ground-truth communities using the LFR (Lancichinetti Fortunato Radicchi) benchmark [33]. The parameter settings of the LFR benchmark are provided below.

The number of nodes in the network was set to n=10,000. The average node degree of the network was set to $\bar{d}=5$. The minimum and maximum size of the community were set to ${\left|C\right|}_{\min}=5$ and ${\left|C\right|}_{\max}=500$, respectively. The overlap degree of each overlapping node was set to $O_{m}=2$. The number of overlapping nodes in the network was set to $O_{n}=500$. The mixing parameter μ was set to {0.1:0.1:0.8}, that is, the value of μ varied within the range from 0.1 to 0.8 with a span of 0.1. As μ increases, community boundaries become blurred and communities in the network become less identifiable.

#### 5.2.2. Complex Networks

Complex networks are used to validate the performance of GTIP and traditional community detection methods.

(1) The College Football Network. This network contains 115 nodes and 616 edges, where the nodes in the network represent the football team and the edge between two nodes indicates that there has been a game between the two teams.

(2) The Political Book Network. This network is generated by the sales records of political books on Amazon.com during the president election in the early 21st century, and consists of 105 nodes and 441 edges. The nodes represent the book and the edge represents co-purchasing of books by the same buyers. The network forms three natural communities, “liberal”, “neutral”, and “conservative”.

(3) The Dolphin Family Network. The network consists of two dolphin families with 62 nodes and 159 edges. The nodes in the network represent dolphins and the edge represents the frequency of contact between two dolphins.

#### 5.2.3. Real-World Networks

Real-world networks were used to validate the performance of GTIP and semantic community detection methods. The five semantic-rich real-world networks used in the experiment can be downloaded from https://www.aminer.cn (accessed on 1 August 2022) and https://snap.stanford.edu/data/index.html (accessed on 1 August 2022). (1) Academic Social Network (ASN): this dataset includes paper information, paper citations, author information, and author collaboration, and contains 1,712,433 authors (nodes) and 4,258,615 collaboration relationships (edges).

(2) Youtube social network: Youtube is a video-sharing website where users can establish friendships and create groups. This dataset contains 1,134,890 nodes and 2,987,624 edges.

(3) DBLP collaboration network: the DBLP computer science bibliography provides a comprehensive list of research papers in computer science. This dataset contains 317,080 nodes and 1,049,866 edges.

(4) Amazon product co-purchasing network: this network was collected by crawling the Amazon website. If a product *i* is frequently co-purchased with product *j*, the graph contains an undirected edge from *i* to *j*. This dataset contains 334,863 nodes and 925,872 edges.

(5) Enron email network: this dataset was originally made public and posted to the web by the Federal Energy Regulatory Commission during its investigation of Enron; it contains 36,692 nodes and 183,831 edges.

### 5.3. Parameter Analysis

#### 5.3.1. Analysis on the Influence Range of Percolation

We use the parameter jump to represent the influence range of percolation, which can affect the aggregation of nodes inside communities.

In non-artificial network experiments (Table 5 and Table 6), when jump increases the number of detected communities decreases and the quality (EQ and SQ) of communities declines, especially when jump>3. According to Equation (Equation 29), the source point has a great influence on the nodes within three hops. Beyond this range there is uncertainty in percolation, which leads to the fragmentation of the community and reduces the quality of the community structure. In comparison, the decay rate of SQ is slightly faster than that of EQ. Comparing with Equations (Equation 34) and (Equation 35), changes in percolation range are more likely to affect the similarity between nodes within the community than that community’s proportion of internal and external links.

In artificial network experiments (Table 7), the performance of GTIP varies with parameter μ. As μ increases, communities in the network become less identifiable and the NMI score gradually decreases. The performance of GTIP continues to decrease rapidly when j>3. In contrast to non-artificial network experiments, the difference in NMI score for j=1, j=2, and j=3 is not significant. One possible reason for this is that the link distribution of the non-artificial network is relatively uniform, which decreases the difference in node influence within three hops.

In summary, the performance of GTIP is weak when Jump=3, and the percolation is ineffective when Jump>3. Without loss of generality, we set Jump=3 in the following experiments.

#### 5.3.2. Analysis on the Number of Topics

The number of topics (#Topics) in a document collection *T* can affect the size of the base of the semantic space; therefore, we verified the change in community quality when the number of topics was T=1,2,...,20.

The experiment results are shown in Figure 4, Figure 5 and Figure 6. It can be seen that when #Topics ranges from 0 to 8, the quality (EQ, SQ and NMI) of communities increases exponentially. When #Topics ranges from 8 to 12, the quality of communities tends to be stable. When #Topics ranges from 12 to 20, the quality of communities decreases rapidly. The reason for this is that when #Topics increases, the difference in the semantic space coordinate of each node becomes larger, which increases the possibility of community division. In this experiment, EQ, SQ, and NMI reach the optimal value when the number of topics is around 10. In addition, the SQ values of community structures are higher in networks with obvious topic attributes. For example, the topics in the Enron email network mostly focus on finance, stock price, and energy transportation, which makes the community have strong topic consistency. To better demonstrate the performance of our algorithm, we set #Topics =10 in the following experiments.

### 5.4. Experimental Results on Artificial Networks

We executed eleven community detection algorithms on LFR artificial networks and recorded the NMI values. From Table 8, it can be seen that complex network community detection methods (GN, FN, LFM, and COPRA) have lower NMI values, while the NMI values slowly decreases when μ becomes large. In comparison, COPRA performs better and remains effective in mining the community structure when the community boundaries are blurred (μ=0.6,0.7 and 0.8). As the community boundaries become clearer, the performance of the semantic community discovery algorithm improves. When μ=0.4 and 0.5, ACQ and CUT have a higher NMI value. GTIP and DEEP perform better when μ=0.1,0.2 and 0.3. However, because DEEP requires a large number of ground-truth communities as samples, its NMI decays faster as μ grows larger. In comparison, GTIP has better performance. The reason for this is that when the community boundary is clearer, node cohesiveness and central tendency are stronger, which is more consistent with the community generation principle of GTIP.

### 5.5. Experimental Results on Complex Networks

We chose the Football, Books, and Dolphins networks as the experimental datasets. The algorithms used for the comparison included GN [6], FN [7], LFM [33], and COPRA [34]. GN and FN are non-overlapping community detection algorithms, while LFM and COPRA are overlapping community detection algorithms. We compared the EQ and SQ of each algorithm on the three complex networks described in Section 5.2.2.

Table 9 shows the EQ and SQ score of each algorithm. GN and FN discover communities by cutting edges and if communities do not overlap, their EQ values are lower. LFM and COPRA aim to increase the proportion of internal and external links of the community, therefore, the EQ value of the two algorithms is higher than that of GTIP (5.229% higher on average). The goal of GTIP is semantic similarity among nodes in the community, therefore, the SQ value of GTIP is higher than the other four algorithms (27.153% higher on average). The COPRA algorithm has the highest EQ value in the experiment; its SQ value, however, is lower than GTIP algorithm (8.184% lower on average). In general, traditional non-semantic community detection algorithms have high performance in mining communities based on topology structure and poor performance in community detection with rich semantic information.

Horizontal comparison shows that the EQ value of the classical community detection algorithms is higher than the SQ value (10.169% higher on average). COPRA and GTIP show good performance on complex networks. Both of them discover communities based on information diffusion, which indicates that accurately simulating the interaction behavior of social individuals is an effective way to detect communities with tight structure and semantic cohesion.

### 5.6. Experimental Results on Real-World Networks

In this section, we compare GTIP with seven semantic community detection algorithms: CUT [35], TURCM [36], LCTA [37], ACQ [21], DEEP [28], BTLSC [29], and SCE [14]. We used the five real-world networks described in Section 5.2.3 as the experiment data; the results are shown in Table 10 and Table 11.

BTLSC and SCE have better performance on ASN and Youtube networks. For example, in the EQ comparison experiment, BTLSC and SCE outperform GTIP by 0.294% and 11.233%, respectively. In the SQ comparison experiment, BTLSC and SCE outperform GTIP by 2.369% and 12.384%, respectively. On the DBLP, Amazon, and Enron networks, GTIP has a definite performance advantage. In the EQ and SQ comparison experiment, GTIP outperforms the other algorithms by an average of 18.386% and 19.973%, respectively. The reason for this is that the nodes in these three networks generally have a high propensity for topics. Taking the Enron network as an example, Figure 7 depicts the word clouds of the Enron network. It can be seen that the network has a strong topic concentration containing six distinct topics, which enhances the accuracy of the GTIP algorithm in selecting the source point of percolation. Additionally, in networks with rich semantic information SQ is typically lower than EQ. The reason for this is that in a semantic social network, although two users may focus on the same topic, different sentiment tendencies concerning the topic can lead to a split in the community.

## 6. Conclusions

This paper proposes GTIP, a semantic community detection method based on topic influence percolation. First, we modeled topic propagation in semantic social networks as the flow of a fluid through porous media based on percolation mechanics, then constructed a partial differential equation to solve the percolation intensity of topic influence. Second, based on game theory, the rules of accepting and forwarding topics were formulated to maximize the benefits of users and achieve Nash equilibrium. Finally, a semantic community was generated based on the seed expansion process.

We conducted experiments on artificial networks, complex networks, and semantic social networks. Our results show that when community boundaries are obvious and the corpus is rich, the modularity and NMI scores of GTIP are significantly better than other comparison algorithms. This shows that GTIP can capture the structural density and semantic cohesion of the network and has a high performance advantage in networks with high topic concentration.

In fact, users have different emotional perceptions of different topics, and even if we gather users with similar topics into one community, the community has the potential to split. In future work, we intend to integrate the sentiment attributes into the base of the semantic space in order to improve the structural stability of the detected communities.

## Figures and Tables

**Figure 2 entropy-24-01274-f002:**
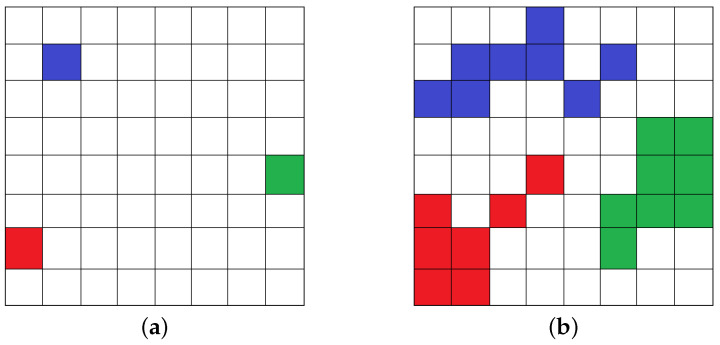
The percolation process of the fluid. (**a**) Initial percolation state. (**b**) Percolation state after time *t*.

**Figure 3 entropy-24-01274-f003:**
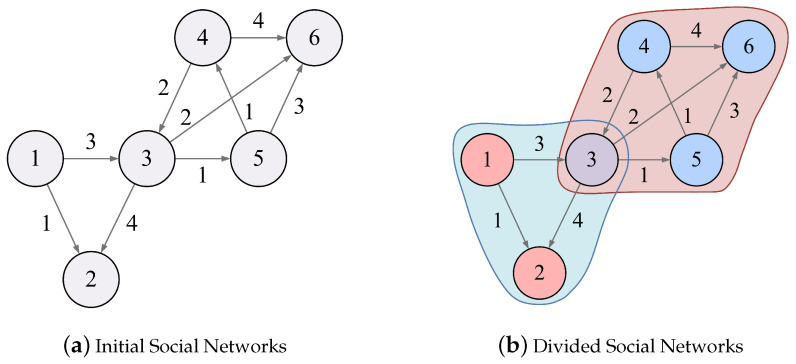
Community detection with GTIP algorithm.

**Figure 4 entropy-24-01274-f004:**
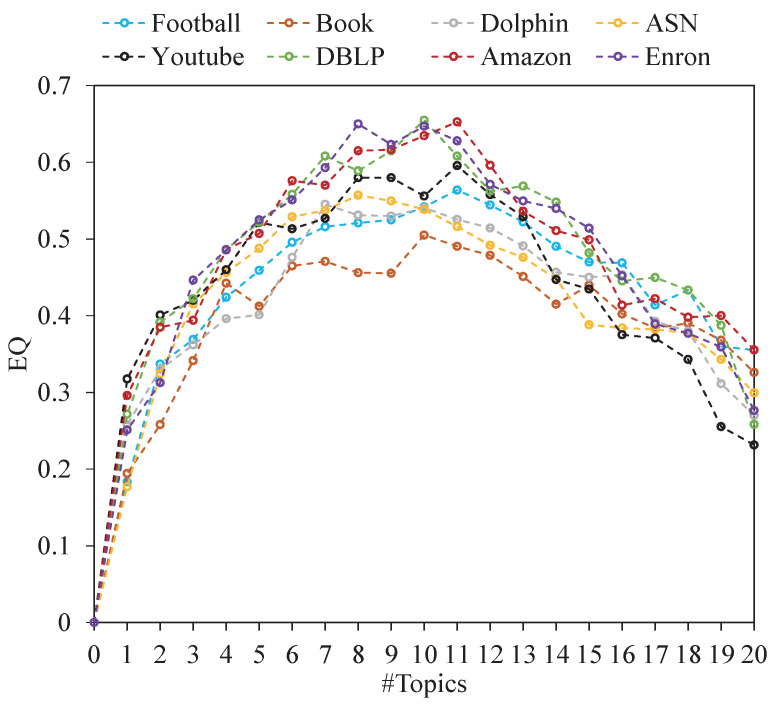
The EQ value on non-artificial networks with #Topics range from 1 to 20.

**Figure 5 entropy-24-01274-f005:**
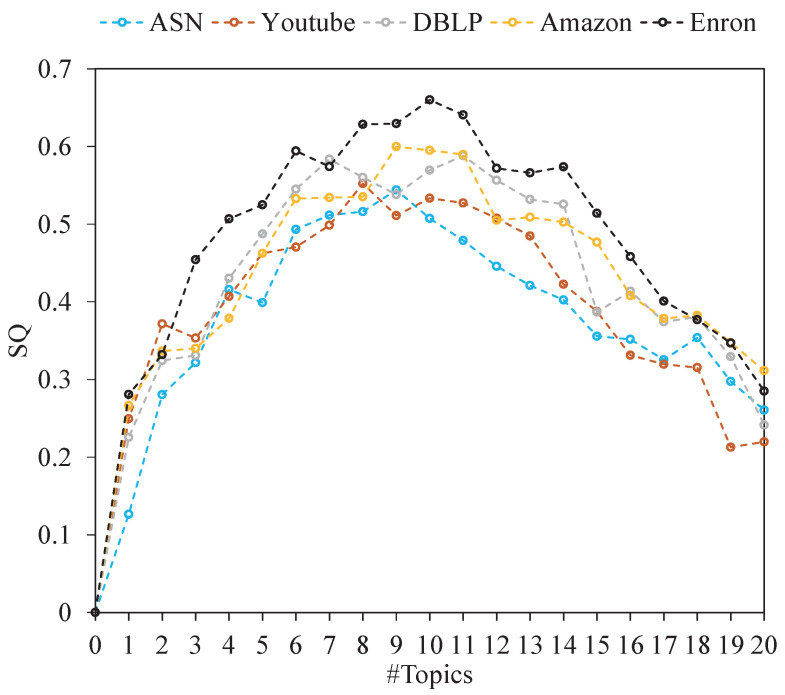
The SQ value on non-artificial networks with #Topics range from 1 to 20.

**Figure 6 entropy-24-01274-f006:**
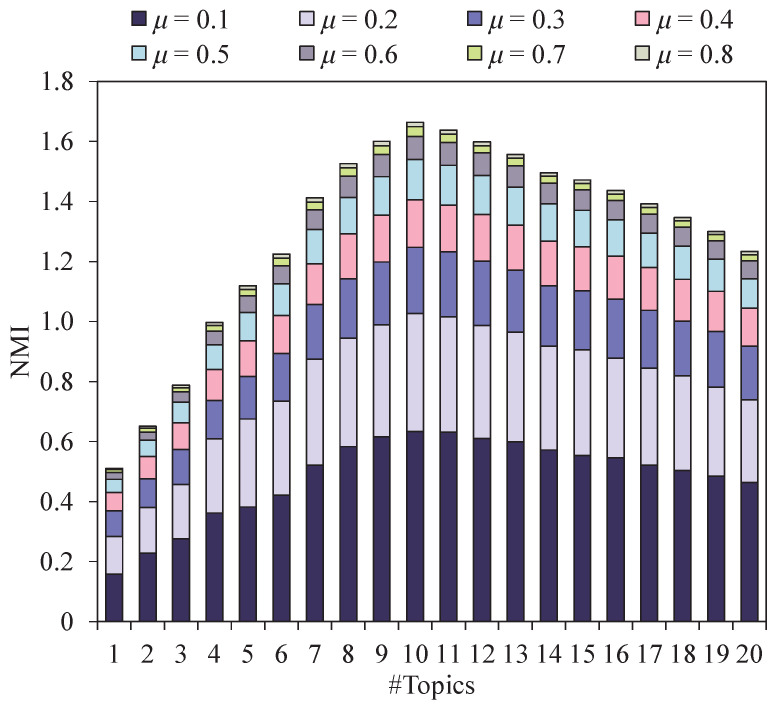
The NMI score on non-artificial networks with #Topics range from 1 to 20.

**Figure 7 entropy-24-01274-f007:**
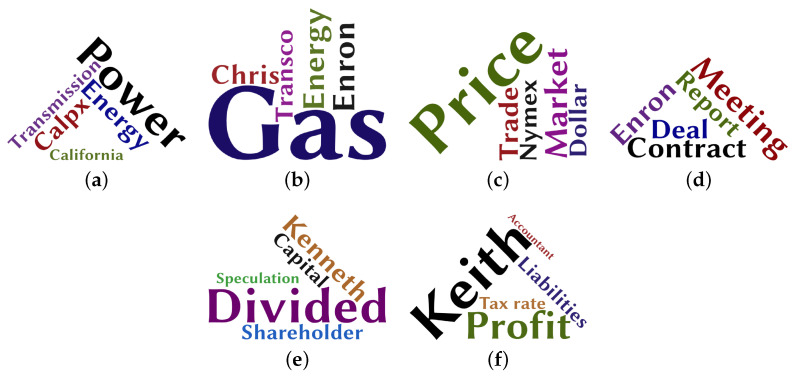
Word clouds of six topics on Enron network: (**a**) California power, (**b**) Gas_trans, (**c**) Trading, (**d**) Deals, (**e**) Stock, (**f**) Finance.

**Table 3 entropy-24-01274-t003:** Description of notations.

Notation	Description
*S*	The topic influence percolation strength
λ	The dimension of topic influence
γ	The sphere volume
$Z_{i\to j}$	The topic propagation space coordinate
*D*	The hops between the source point and the affected nodes
$\eta_{z}$	The percolation coefficient of topic propagation
$\kappa_{0}$	The initial topic influence value of the source point

**Table 4 entropy-24-01274-t004:** The topic influence score of nodes in $G_{a}$.

Node ID	Topic Influence Score
1	11.51
2	25.44
3	12.99
4	10.13
5	11.51
6	28.41

**Table 5 entropy-24-01274-t005:** The EQ value on non-artificial networks with Jump range from 1 ro 6.

Jump	Football	Book	Dolphin	ASN	Youtube	DBLP	Amazon	Enron
1	0.5125	0.4847	0.4656	0.4251	0.3621	0.7015	0.8982	0.8322
2	0.5531	0.5022	0.4738	0.4315	0.3625	0.7088	0.9011	0.8414
3	0.4203	0.4291	0.3928	0.4422	0.3638	0.7147	0.9132	0.8543
4	0.2103	0.2159	0.2112	0.2133	0.2191	0.2189	0.2176	0.2102
5	0.0054	0.0098	0.0085	0.0073	0.0025	0.0018	0.0032	0.0047
6	0.0000	0.0000	0.0000	0.0000	0.0000	0.0000	0.0000	0.0000

**Table 6 entropy-24-01274-t006:** The SQ value on non-artificial networks with Jump range from 1 ro 6.

Jump	ASN	Youtube	DBLP	Amazon	Enron
1	0.4206	0.3539	0.7132	0.8149	0.8101
2	0.4142	0.3393	0.6930	0.8273	0.8266
3	0.3968	0.3207	0.6865	0.8076	0.8064
4	0.1074	0.1071	0.1089	0.1101	0.1106
5	0.0032	0.0028	0.0035	0.0048	0.0045
6	0.0000	0.0000	0.0000	0.0000	0.0000

**Table 7 entropy-24-01274-t007:** The NMI value on artificial networks with Jump range from 1 ro 6.

Jump	μ = 0.1	μ = 0.2	μ = 0.3	μ = 0.4	μ = 0.5	μ = 0.6	μ = 0.7	μ = 0.8
1	0.6291	0.3847	0.2264	0.1537	0.1324	0.0759	0.0235	0.0128
2	0.6213	0.3796	0.2201	0.1471	0.1294	0.0712	0.0212	0.0117
3	0.6185	0.3713	0.2165	0.1406	0.1235	0.0673	0.0196	0.0087
4	0.0017	0.0015	0.0014	0.0012	0.0011	0.0008	0.0006	0.0004
5	0.0005	0.0004	0.0003	0.0002	0.0002	0.0001	0.0001	0.0001
6	0.0000	0.0000	0.0000	0.0000	0.0000	0.0000	0.0000	0.0000

**Table 8 entropy-24-01274-t008:** The NMI value on artificial networks.

Algorithms	μ = 0.1	μ = 0.2	μ = 0.3	μ = 0.4	μ = 0.5	μ = 0.6	μ = 0.7	μ = 0.8
GN	0.1823	0.0836	0.0551	0.0179	0.0064	0.0037	0.0006	0.0000
FN	0.1912	0.0861	0.0573	0.0167	0.0066	0.0029	0.0003	0.0000
LFM	0.2107	0.1684	0.1357	0.1122	0.0638	0.0311	0.0103	0.0043
COPRA	0.5158	0.3988	0.3342	0.2801	0.2466	0.2272	0.2168	0.1707
CUT	0.4758	0.3864	0.3561	0.2806	0.2653	0.2232	0.1837	0.1332
TURCM	0.5066	0.4213	0.3722	0.2313	0.1892	0.1534	0.1108	0.0606
LCTA	0.3851	0.3762	0.3097	0.2885	0.2204	0.1818	0.1390	0.0923
ACQ	0.4344	0.4099	0.3768	0.3159	0.2351	0.2111	0.1792	0.1017
DEEP	0.5932	0.4645	0.3536	0.2837	0.1963	0.0992	0.0103	0.0008
BTLSC	0.5022	0.3269	0.2851	0.2022	0.1733	0.1431	0.1005	0.0520
SCE	0.4224	0.3861	0.3236	0.2619	0.2052	0.1337	0.0838	0.0153
GTIP	0.6185	0.4713	0.4065	0.2406	0.1235	0.0673	0.0196	0.0087

**Table 9 entropy-24-01274-t009:** Performance comparison with traditional community detection algorithms.

Algorithm	Criteria	Football	Book	Dolphin
GN	EQ	0.2977	0.3084	0.3165
	SQ	0.2821	0.2927	0.3002
FN	EQ	0.2876	0.2988	0.3153
	SQ	0.2774	0.2831	0.3032
LFM	EQ	0.4207	0.4266	0.4137
	SQ	0.3831	0.3515	0.3604
COPRA	EQ	0.4858	0.4672	0.4003
	SQ	0.4115	0.3728	0.3948
GTIP	EQ	0.4203	0.4291	0.3928
	SQ	0.4326	0.4364	0.4066

**Table 10 entropy-24-01274-t010:** The EQ value on real-world networks.

Networks	CUT	TURCM	LCTA	ACQ	DEEP	BTLSC	SCE	GTIP
ASN	0.2466	0.3867	0.3580	0.3458	0.3623	0.4435	0.4295	0.4422
Youtube	0.3278	0.3445	0.4362	0.3287	0.4494	0.4224	0.5159	0.4638
DBLP	0.6048	0.6413	0.6082	0.4846	0.5846	0.6520	0.6953	0.7147
Amazon	0.7221	0.8128	0.7090	0.6940	0.8017	0.8043	0.8910	0.9132
Enron	0.6436	0.7405	0.6512	0.6332	0.8013	0.6712	0.8261	0.8543

**Table 11 entropy-24-01274-t011:** The SQ value on real-world networks.

Networks	CUT	TURCM	LCTA	ACQ	DEEP	BTLSC	SCE	GTIP
ASN	0.2012	0.3357	0.3106	0.2977	0.3212	0.4062	0.3862	0.3968
Youtube	0.2918	0.3014	0.3931	0.2856	0.4041	0.3762	0.4728	0.4207
DBLP	0.5766	0.6153	0.5841	0.4564	0.5639	0.6262	0.6723	0.6865
Amazon	0.6127	0.7072	0.6068	0.5833	0.6925	0.7011	0.7854	0.8076
Enron	0.5936	0.6957	0.6012	0.5831	0.7534	0.6233	0.7761	0.8064

## Data Availability

The publicly available datasets analyzed for this study can be found at (https://www.aminer.cn accessed on 1 August 2022) and (https://snap.stanford.edu/data/index.html accessed on 1 August 2022). Further inquiries can be directed to the corresponding author.

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
