# Peer review of "GTIP: A Gaming-Based Topic Influence Percolation Model for Semantic Overlapping Community Detection"

_entropy, 2022, doi:10.3390/e24091274_

Round 1

Reviewer 1 Report

In the paper, the authors propose Gaming-based Topic Influence Percolation model (GTIP) for semantic overlapping community detection.
This article is recommended for publication with the following minor modifications:
- More detailed should be added in the further discussions and conclusion section.
- As per the Turnitin software, the similarity index is 23% can be improved.
- The abbreviations should be defined at the first place of its use in the article. Please check and do the needful.

Author Response

Dear reviewer,

Thank you for your comments on our manuscript. Please see the attachment for our revision instructions.

Reviewer 2 Report

The paper proposes a user topic influence propagation model based on percolation theory and a game theoretic approach based on NE to generate communities. Experiments on real social networks show that the proposed method has a high semantic modularity in social networks with rich semantic attributes. The algorithm converges in short times without additional training.

The topic of the paper is interesting and could have practical merit. It combines different research approaches and provides extensive evaluation.

A fundamental problem of the paper holistically is that the content is not presented appropriately. There are problems with the text, and the latter is sometimes not as explanatory as it should be.

The notions of topic, keywords, words have not been explained adequately prior to their use. Perhaps a simple concrete example regarding words topics and keywords should be included to make them more specific.

In the beginning of section 3.1, the introduction of percolation is unsuccessful. Figure 2 and the relevant lines 186-188 are fuzzy. Details on the figure are not provided. For instance, what do the colors stand for, what are the evolutionary mechanics, how the figure evolves from 2(a) to 2(b), etc. More elaborate and progressive explanation of the setting and evolution is needed to demonstrate properly the operation of percolation.

In lines 197-199 the use of the percolation model assumes uniform probability of diffusion among neighboring sites (nodes). However, this is not certain when it comes to information diffusion among people, who might exhibit preference, topical interest, etc. A convincing justification should be provided.

In line 220, what is meant by “size of solid sphere”? Do the authors refer to sphere volume?

Equation (8) just emerges without prior explanation. Please provide adequate reasoning.

In line 267, the authors refer to the 3σ principle of normal function. Although they refer to a well-known property regarding the concentration of mass of the normal distribution around its mean, there is no such standard terminology as “3σ principle of normal function”. It is suggested to simply refer to the property, without verbal callouts.

Regarding eqs. (21) and (22), I don't see a meaning in (22). Eq. (21) defines the corresponding relationship in a causal manner, while eq. (22) essentially reverses such causality.

In lines 356-357, it is stated that “we assume that there is no percolation between v_i and v_j when m_i \times m_j < 0.2”. How is this justified?

Regarding eqs. (34) and (35), they contain some standard quantities such as node degree, total network degree, etc. Typically, some standard symbols are used for them, but here they are all modified to non-meaningful symbols. It would be better to maintain the frequently used symbols for frequently used quantities.

The abstract contains too specific/technical terminology, which is inappropriate for its purpose. It is necessary to simplify it, focusing on an intuitive explanation of the approach, rather than technical details. Also, in lines 12-13, the abstract should be specific regarding the comparison with other approaches, specifying them explicitly for ease of the authors’ understanding.

In lines 22-27, the authors need to provide references for the data included.

The LDA vector mentioned in the beginning of section 2.1, line 133, should be explained in more detail.

In lines 141-148 new symbols are introduced. They should be defined prior to their use in the specific part of the paper. Furthermore, more elaboration is needed for the corresponding part of the paper.

A notation table is needed. The paper uses quite many symbols and it would be better to collect them all in a table for quick reference.

There is no acronym list, typically included in MPDI paper.

Author Response

(The authors gave the same response as above.)

Round 2

Reviewer 2 Report

My comments have been addressed. A final proofreading is needed to make sure there are no typos left in the manuscript.